# Modulation of the Gut Microbiota by the Plantaricin-Producing *Lactiplantibacillus plantarum* D13, Analysed in the DSS-Induced Colitis Mouse Model

**DOI:** 10.3390/ijms242015322

**Published:** 2023-10-18

**Authors:** Katarina Butorac, Jasna Novak, Martina Banić, Andreja Leboš Pavunc, Nina Čuljak, Nada Oršolić, Dyana Odeh, Jana Perica, Jagoda Šušković, Blaženka Kos

**Affiliations:** 1Laboratory for Antibiotic, Enzyme, Probiotic and Starter Culture Technologies, Department of Biochemical Engineering, Faculty of Food Technology and Biotechnology, University of Zagreb, Pierottijeva 6, 10000 Zagreb, Croatia; katarina.butorac@pbf.unizg.hr (K.B.); jasna.novak@pbf.unizg.hr (J.N.); mmarijanovic@pbf.unizg.hr (M.B.); andreja.lebos.pavunc@pbf.unizg.hr (A.L.P.); nina.culjak@pbf.unizg.hr (N.Č.); jperica@pbf.hr (J.P.); jsusko@pbf.hr (J.Š.); 2Department of Animal Physiology, Faculty of Science, University of Zagreb, Rooseveltov trg 6, 10000 Zagreb, Croatia; nada.orsolic@biol.pmf.hr (N.O.); dyana.odeh@biol.pmf.hr (D.O.)

**Keywords:** bacteriocin, plantaricin, *Lactiplantibacillus plantarum*, probiotic, inflammatory bowel disease, colitis, intestinal microbiota

## Abstract

*Lactiplantibacillus plantarum* D13 shows antistaphylococcal and antilisterial activity, probably due to the synthesis of a presumptive bacteriocin with antibiofilm capacity released in the cell-free supernatant (CFS), whose inhibitory effect is enhanced by cocultivation with susceptible strains. An in silico analysis of the genome of strain D13 confirmed the *pln* gene cluster. Genes associated with plantaricin biosynthesis, structure, transport, antimicrobial activity, and immunity of strain D13 were identified. Furthermore, the predicted homology-based 3D structures of the cyclic conformation of PlnE, PlnF, PlnJ, and PlnK revealed that PlnE and PlnK contain two helices, while PlnF and PlnJ contain one and two helices, respectively. The potential of the strain to modulate the intestinal microbiota in healthy or dextran sulphate sodium (DSS)-induced colitis mouse models was also investigated. Strain D13 decreased the disease activity index (DAI) and altered the gut microbiota of mice with DSS-induced colitis by increasing the ratio of beneficial microbial species (*Allobaculum, Barnesiella*) and decreasing those associated with inflammatory bowel disease (*Candidatus Saccharimonas*). This suggests that strain D13 helps to restore the gut microbiota after DSS-induced colitis, indicating its potential for further investigation as a probiotic strain for the prevention and treatment of colitis.

## 1. Introduction

The gut microbiota represents a complex ecosystem that inhabits the digestive tract and lives in symbiosis with the host. Dysbiosis of the gut microbiota is associated with many diseases, primarily those related to the disruption of intestinal dysfunction, such as inflammatory bowel disease (IBD), represented by ulcerative colitis and Crohn’s disease, and irritable bowel syndrome (IBS) [1,2]. The link between the composition of the intestinal microbiota and the development of disease has been extensively studied [1,2]. This has led to the development of microbial-based and other strategies to alter its composition, such as manipulation with prebiotics (mainly dietary fibers), probiotics, synbiotics, parabiotics, and/or postbiotics and equivalent synthetic products that positively influence the immunological response and inflammation [3]. *Lactiplantibacillus plantarum* (formerly named *Lactobacillus plantarum*) (*Lb. plantarum*) is one of the most extensively studied lactic acid bacteria (LAB), ubiquitous in the various microbiomes of fermented foods and in the human microbiome [4,5,6]. In addition, certain strains isolated from human saliva or the gastrointestinal tract (GIT) are considered important for health-promoting host–microbe interactions [7]. In both aspects of application, either as a functional starter or probiotic culture, the potential biosynthesis of bacteriocins is attractive as it ensures a competitive advantage and eliminates undesirable contaminating strains that may be present in food or gut microbiomes. Bacteriocin-producing probiotic strains have the ability to modulate the gut microbiota by conferring resistance to pathogen colonisation. Namely, these beneficial bacteria can inhibit the colonisation of pathogenic bacteria by synthesising antimicrobials such as lactic acid, thus lowering the luminal pH, or by producing bacteriocins, thus inhibiting pathogens, as well as through competing for intestinal sites and nutrients, forming coaggregates and finally eliminating undesirable species. This is the reason that bacteriocins are not only recognised as antimicrobial peptides but also as signalling or colonising molecules. These functions, which may be synergistic, are the basis for the establishment of a producer strain in the GIT, its immunomodulatory activity or its direct competitive exclusion of pathogens, and its ability to biomodulate the composition of the gut microbiota [8]. Another functional aspect of bacteriocins is their activity against biofilms of foodborne pathogens and multidrug-resistant bacteria, which is important in the context of food safety and antibiotic resistance, as resistant bacteria in biofilms can withstand a thousand-fold higher concentration of antibiotics compared to planktonic cells [9]. Therefore, bacteriocins can be used as potential natural alternatives to effectively inhibit biofilm formation and disrupt already formed biofilms, either alone or in combination with drugs other than antibiotic antagonists [10]. Advances in the characterisation of bacteriocins have sparked interest in the industrial application of these molecules either as biopreservatives or as new-generation antimicrobials [11]. The aim of this study was to examine, among selected autochthonous *Lb. plantarum* strains, those with the most pronounced antibacterial activity and to focus on their potential for the synthesis of bacteriocins. Next, we explored the potential of bacteriocin-producing *Lb. plantarum* to beneficially modulate the composition of the gut microbiota in mice with DSS-induced colitis.

## 2. Results

### 2.1. Selection of Lb. plantarum Strains with Potential Plantaricin Production

The potential bacteriocin activity of twelve *Lb. plantarum* strains was tested in vitro by an agar spot assay against the common Gram-positive pathogens *S. aureus* ATCC^®^ 25923^™^ and *L. monocytogenes* ATCC^®^ 19111^™^. The analysed strains of *Lb. plantarum* showed antistaphylococcal and antilisterial activity. The inhibitory effect was more pronounced against *Staphylococcus aureus* ATCC^®^ 25923^™^ than against *Listeria monocytogenes* ATCC^®^ 19111^™^, since all strains of *Lb. plantarum* displayed strong antistaphylococcal inhibition, while the antilisterial activity was characterised as medium and strong, depending on the strain (Appendix A). Based on the results, six *Lb. plantarum* strains (D13, M5, SF15C, ZG1C, M92C, L4) with the most prominent antibacterial activity were selected for the further characterisation of potential plantaricin production. Since the bacteriocins of the class II, which includes plantaricins, are active in both acidic and neutral environments but can be inactivated by proteases, such as trypsin from the pancreas and pepsin from the stomach, cell-free supernatants (CFSs) of *Lb. plantarum* strains were exposed to proteinase K, pepsin, and pancreatin and a temperature of 100 °C for 30 min. These specific treatments resulted in a partial reduction in the antistaphylococcal and antilisterial activity of *Lb. plantarum* D13 and SF15C, respectively (*p* < 0.05), indicating the proteinaceous nature of the antimicrobial component (Appendix A), whereby the D13 strain displayed the stronger inhibition. The protein nature of the antimicrobial compound in the CFSs of the M92C strain was assumed, since the proteolytic enzymes applied partially abolished the inhibitory activity against both test microorganisms (*p* < 0.05). Furthermore, pancreatin treatment significantly (*p* < 0.05) impacted the decrease in the antibacterial activity of the CFSs of strains M5 and ZG1C against *L. monocytogenes* ATCC^®^ 19111^™^, and, in the same manner, pepsin treatment reduced the antimicrobial potential of the CFS of strain ZG1C.

Next, the selected *Lb. plantarum* strains were screened for the presence of *pln* loci by PCR using gene-specific primers. According to the results, the plantaricin-related genes *pln*A, *pln*EF, and *pln*J were detected with an expected amplicon size of 450 bp, 428 bp, and 475 bp, respectively (Appendix A). The genes *pln*NC8, *pln*S, and *pln*W were not amplified and the negative control did not result in amplification, confirming the absence of contamination or non-specific amplification. Based on the results of the microbiological and genomic analysis, the strain *Lb. plantarum* D13 was selected for the further characterisation of plantaricin biosynthesis. The molecular identity of the PCR amplicons of strain D13 was confirmed by sequencing and aligned with the nucleotide sequences deposited in NCBI. The sequences of the amplicons showed a high level of sequence identity (>98.2%), indicating that the genome harbours a *pln* gene cluster (Appendix A).

One approach to characterising bacteriocins is to stimulate their synthesis by coculturing a producer strain with a sensitive bacterium. Therefore, *Lb. plantarum* D13 was cocultivated with *L. monocytogenes* ATCC^®^ 19111^™^ and *S. aureus* ATCC^®^ 25923^™^ to assess the potential of coculture-inducible plantaricin production. After 24 h, the number of culturable cells of *L. monocytogenes* ATCC^®^ 19111^™^ decreased significantly from 8.93 (±0.13) to 5.97 (±0.25) log (CFU/mL). The inhibitory effect was also present, although less pronounced, when D13 was cocultured with *S. aureus*^®^ 25923^™^ (from 9.05 (±0.07) to 7.39 (±0.12) log (CFU/mL)) (Figure 1a). During the cocultivation experiments, the inhibitory activity of D13 was determined by the agar spot assay method. The induction of plantaricin activity was enhanced during the exponential growth phase in coculture and persisted throughout the stationary growth phase, compared to the activity when the D13 strain was grown in monoculture (Figure 1b). The antimicrobial activity of the acids produced by the metabolism of the *Lb. plantarum* D13 strain was eliminated by monitoring the pH values during the cocultivation, which corresponded to the values measured in the monoculture of the test microorganisms, and additionally by testing their inhibition using the determined lactic acid concentrations in coculture (8.1 and 6.3 g/L with *L. monocytogenes* ATCC^®^ 19111^™^ and *S. aureus* ATCC^®^ 25923^™^, respectively) at the end of the experiment. Considering the monitored pH values and the stronger antimicrobial activity of CFSs compared to lactic acid, it can be speculated that the inhibitory effect of the *Lb. plantarum* D13 strain during cocultivation is the result of possible plantaricin activity.

### 2.2. Putative Plantaricins as Agents against Biofilm Formation

Foodborne biofilms are of particular concern to the food industry, and *S. aureus* and *L. monocytogenes* are among the most significant biofilm formers [10]. Here, we investigated whether the CFS of *Lb. plantarum* D13 could inhibit biofilm formation by *L. monocytogenes* ATCC^®^ 19111^™^ and *S. aureus* ATCC^®^ 25923^™^. When a bacterial suspension of *L. monocytogenes* ATCC^®^ 19111^™^ was treated simultaneously with the CFS of *Lb. plantarum* D13, biofilm formation was significantly (*p* < 0.001) reduced compared to the control, i.e., the bacterial suspension alone, and the inhibition rate of biofilm formation was 64.59 (±6.94)% (Figure 2). A similar effect was obtained against *S. aureus* ATCC^®^ 25923^™^, where the inhibition rate of biofilm formation was 69.58 (±14.97)%. Therefore, it can be speculated that the putative plantaricins in CFS possesses a certain ability to inhibit the biofilm formation of these two pathogen representatives. Given the demonstrated antibiofilm and antimicrobial activity of *Lb. plantarum* D13, we can suggest its potential ability to eradicate an already formed biofilm in the microenvironment against susceptible bacterial species (Figure 1 and Figure 2).

### 2.3. In Silico Characterisation of Plantaricin-Related Genes

The whole genome of *Lb. plantarum* D13 was sequenced to confirm the identification of the strain and to identify potential bacteriocin cluster genes that may be related to the observed antibacterial phenotype. The genome size of D13 is 3.34 Mb and it has GC content of 44.4%. The threshold in finding the corresponding genes was set at a region size of 16,000 bp, 15 regions, and an e-value of 1 × 10^−20^ for the selection of pinned coding sequences. In total, 22 genes related to planataricin synthesis were found in contigs 7, 12, and 14. The genes showed high similarity to those of the *Lb. plantarum* WCFS1 genome, with the exception of the immunity protein PlnP, which is located upstream of the *pln*MNOP operon on contig 14. The *Lb. plantarum* D13 genome contains five plantaricin-specific operons on contigs 7 and 12, including the regulator operon *pln*ABCD, the plantaricin operons *pln*EFI, *pln*JKLR, and *pln*MNOP, and the transport operon *pln*GHTUVW, which are required for the production of the class IIb bacteriocins, plantaricin *pln*EF, *pln*JK, and *pln*A (Figure 3a). Genes associated with the biosynthesis, structure, transport, and antimicrobial activity of bacteriocins and the immunity of the producer strain are listed in Appendix A. The alpha helices in the bacteriocin structure determine its function. Thus, the predictions of the cyclic conformation of PlnE, PlnF, PlnJ, and PlnK were performed in silico, according to previous structural studies of the respective plantaricins available in the AlphaFold Protein Structure Database, with a threshold of 100% sequence similarity. Since their structures share 100% amino acid identity with the respective peptides of other strains with characterised plantaricin antimicrobial activity, it is presumed that their function could be similar. According to the resulting homology-based three-dimensional structures, PlnE and PlnK contain two helices, while PlnF and PlnJ contain one and two helices, respectively. HeliQuest analyses of the α-helices of the amino acid sequences of the putative two-peptide plantaricins PlnEF and PlnJK revealed the presence of hydrophobic residues of PlnF, PlnJ (Helix 1), and PlnK (Helix 1 and 2), which are in line with the hydrophobicity, hydrophobic moment, and net charge disclosed (Figure 3b).

### 2.4. The Impact of Lb. plantarum D13 on the Composition of the Gut Microbiome of Mice with DSS-Induced Colitis

Interventional strategies to restore dysbiosis in IBD and to treat it suggest the use of probiotics, of which the use of *Lactobacillus* spp. is highly recommended. *Lb. plantarum* D13 has previously shown significant aggregation capacity and the ability to adhere to the extracellular matrix proteins collagen and laminin, but also to the Caco-2 cell line [12], which is important for probiotic activity.

Therefore, and considering its plantarcin production, we investigated the ability of plantaricin-producing *Lb. plantarum* D13 to alter the composition of the gut microbiota in a mouse model of IBD, particularly in DSS-induced colitis in mice. After the induction of colitis and during the administration of the strain *Lb. plantarum* D13, the disease activity index (DAI) score was determined to monitor the development of colitis and its severity. After DSS-induced colitis, the DAI score was 8.00 ± 1.09 on the sacrifice day, whereas it decreased to 6.00 ± 1.20 after the administration of strain D13, implying a possible positive influence of strain *Lb. plantarum* D13 on the slowing of disease progression and possible recovery (Figure 4a,b). Moreover, a difference in weight loss was observed between the control group with DSS-induced colitis and the group treated with the *Lb. plantarum* D13 strain (9.09 (±0.63)% vs. 6.25 (±0.41)%).

In order to determine whether *Lb. plantarum* D13 improves tissue regeneration and reduces inflammatory cell infiltration, distal colon tissue was analysed to evaluate ulcerative lesions through histopathological score analysis. Before the administration of *Lb. plantarum* D13, haematoxylin and eosin (HE) analysis revealed partially pronounced mucus hypersecretion with a moderately preserved epithelium and glandular architecture. Intramucosal multiplied granulation tissue with more abundant lymphocytic infiltration and foci of high-grade dysplasia were also visible (Figure 4b). The group treated with *Lb. plantarum* D13 showed slightly less mucosal damage. Foci of superficial ulceration were visible, but the architecture of the crypts within the intestinal mucosa was maintained and initial ischemic changes were visible. Visibly increased granulation (inflammatory tissue) was seen in the area of the mucosa with moderately abundant lymphocytic infiltrates. The application of *Lb. plantarum* D13 reduced inflammatory cell infiltration in DSS-induced colitis but did not lead to complete mucosal recovery (Figure 4a). The histological score before the administration of *Lb. plantarum* D13 was 16.17 ± 1.83 points (range 13–18), while, in the DSS group treated with *Lb. plantarum* D13, it was 12.5 ± 2.25 (*p* < 0.05) (range 10–16).

In addition, healthy mice were fed the D13 strain, and 16S rRNA gene analysis of the gut microbiota of the healthy group and the DSS-induced colitis group identified a total of 10 different phyla, 133 bacterial genera, and 226 bacterial species. At the phylum level, *Verrucomicrobia*, *Bacteroidetes*, *Firmicutes*, *Candidatus saccharibacteria*, *Proteobacteria*, and *Actinobacteria* dominated, accounting for more than 99% of the total microbial composition (Figure 4c,d). In the DSS-induced colitis group, the relative abundance of *Bacteroidetes* was significantly (*p* < 0.01) reduced (23.81 (±2.82)% to 3.67 (±1.86)%), while the abundances (*p* < 0.05) of *Firmicutes* (21.66 (±2.97)% to 30.22 (±2.94)%) and *Actinobacteria* (1.46 (±0.40)% to 11.76 (±0.69)%) significantly increased (Figure 4d). After *Lb. plantarum* D13 administration in a model of DSS-induced colitis and in healthy mice, the relative composition of *Firmicutes* (*p* < 0.01) increased from 30.22 (±2.94)% to 47.016 (±0.628)% and 26.06 (±4.10)% to 54.63 (±4.18)%, respectively (Figure 4e,f). After D13 treatment, the level of *Candidatus saccharibacteria* was significantly reduced (*p* < 0.01). In healthy mice, *Lb. plantarum* D13 treatment caused a significant (*p* < 0.05) decrease in the respective levels of *Verrucomicrobia* and *Bacteroidetes* of 15.01 (±6.34)% to 0.42 (±0.03)% and 38.83 (±4.85)% to 11.96 (±2.54)% (Figure 4e). This effect persisted in both cases 5 days after administration with strain D13 (Figure 4e,f). At the species level, a pairwise comparison of the gut microbiota of mice before and after the induction of colitis revealed that treatment with DSS decreased (*p* < 0.05) the abundance of *Barnesiella viscericola*, *Barnesiella intestinihominis*, *Bacteroides acidifaciens*, *Parasutterella excrementihominis*, and *Parabacteroides distasonis*. Conversely, the gut microbiota of the DSS-induced colitis group showed increased relative abundance of *Bifidobacterium pseudolongum*, *Turicibacter sanguinis*, *Lactobacillus reuteri*, *Vallitalea pronyensis*, *Ruminococcus flavefaciens*, *Clostridium ruminantium*, *Staphylococcus saprophyticus*, *Erysipelatoclostridium clostridium cocleatum*, and *Lactobacillus vaginalis* (Figure 5a,b). Of the total 226 species-level operational taxonomic units (OTUs) found in this study, only 148 (65.5%) were found in both groups, while 47 (20.8%) and 31 (13.7%) were found before and after inducing colitis, respectively, supporting the hypothesis of species depletion and consequent reduced biodiversity in the DSS-induced colitis model (Figure 5c). The most abundant species (over 1% of relative abundance) when D13 was administered to healthy mice (Figure 6a) and to mice with DSS-induced colitis (Figure 6b) comprised 26 and 18 species, respectively. The significantly higher relative abundance of *Lb. plantarum* species was observed on day 3 and day 6 of administration in the healthy and DSS-induced mice, respectively. Administration of *Lb. plantarum* D13 enriched the amount (*p* < 0.05) of *Allobaculum*, *Lactococcus lactis*, *Barnesiella viscericola*, and *Olsenella profuse* and depleted *Candidatus Saccharimonas aalborgensis* and *Ruminococcus flavefaciens* in the group of mice with DSS-induced colitis. Treatment with *Lb. plantarum* D13 significantly (*p* < 0.01) increased the abundance of *Lactobacillus*, both in healthy and DSS-induced colitis models, from 8.33 (±2.28)% to 26.25 (±5.28)% and 5.32 (±0.89)% to 14.08 (±1.568)%, respectively.

The microbial diversity of the faecal samples during the administration of *Lb. plantarum* D13 was estimated from the observed OTUs via the Simpson (Figure 6c) and Shannon indices (Figure 6d). During the administration of the *Lb. plantarum* D13 strain, the α-diversity in the form of the Simpson and Shannon indices was significantly increased in the model of healthy mice compared to mice with DSS-induced colitis (*p* < 0.05). Since these index values directly reflect microbial diversity, the results suggest a significant loss of microbial richness in the group of mice with DSS-induced colitis (*p* < 0.05) (Figure 6c,d). This could be due to dysbiosis caused by colitis, where DSS reduced the richness of bacterial species in faecal samples.

## 3. Discussion

*Lactobacillus* spp. produce a range of macromolecules, including bacteriocins, exopolysaccharides (EPSs), surface layer proteins (Slps), and proteinases, which may contribute to the application of producer strains in the development of innovative functional foods or as dietary supplements and even live biotherapeutic products [13,14,15]. Bacteriocins, Slps, and EPSs promote competitive pathogen exclusion, with bacteriocins or biopeptides produced by proteinases exhibiting direct antimicrobial activity [4,16]. The bacteriocins produced by *Lb. plantarum*, termed plantaricins, are generally classified as bacteriocins of the class II and comprise a broad class of bacteriocins with a range of bactericidal/bacteriostatic modes of action [17]. Since these bacteriocins primarily act against Gram-positive bacteria, we aimed to determine whether the antilisterial and antistaphylococcal activity of these strains was related to the bacteriocin-producing phenotype. Therefore, the experimental model was based on testing the antimicrobial activity against *L. monocytogenes*, the most common foodborne pathogen, and *S. aureus*, a model human pathogen commonly isolated as methicillin-resistant *Staphylococcus aureus* (MRSA) [18]. In this study, the phenotype of bacteriocin production in *Lb. plantarum* strains was analysed by considering a group of twelve autochthonous strains. Six of these, which showed significant inhibitory activity against acid-tolerant *L. monocytogens* ATCC^®^ 19111^™^ and *S. aureus* ATCC^®^ 25923^™^, were selected for further PCR analysis of the plantaricin structural genes. PCR-amplified genomic fragments revealed three *pln* genes of interest, which were then sequenced. Furthermore, in vitro studies of CFSs confirmed the presence of a proteinaceous substance, particularly in strain *Lb. plantarum* D13. Strain D13 was previously studied in terms of the techno-functional properties of probiotics, such as survival under stress conditions and colonisation potential (cell surface hydrophobicity; capacity for aggregation and coaggregation; adhesion to mucin, Caco-2 cells, and subepithelial extracellular matrix proteins) [15]. In addition, this strain exhibited caseinolytic activity and, when used as a functional starter culture in the consortium, increased content of peptides with potential bioactive activity was obtained in the dried fresh cheese produced [5,16].

The characterisation of bacteriocins is the most challenging among the antimicrobial substances of LAB, as they are synthesised in very low concentrations. The biosynthesis of bacteriocins is strain-specific and can be optimised via the cultivation conditions and cocultivation with susceptible bacteria. Therefore, to stimulate the antimicrobial activity and synthesis of potential plantaricin, the cocultivation of *Lb. plantarum* D13 with susceptible strains *L. monocytogenes* ATCC^®^ 19111^™^ and *S. aureus* ATCC^®^ 25923^™^ was performed. *Lb. plantarum* D13 showed strong activity against *L. monocytogenes* ATCC^®^ 19111^™^, which is consistent with the antimicrobial spectrum of plantaricins inhibiting *Listeria* spp. The induction of bacteriocinogenic activity was observed during the exponential growth phase. This is consistent with Wu et al. [19], who suggested that the increased bacteriocin activity and expression of genes related to bacteriocins were induced by the cocultivation of *Lb. plantarum* RUB1 with *L. monocytogenes* ATCC^®^ 19111^™^ or
*S. aureus* ATCC^®^ 6538^™^. The antibacterial activity was weaker against *S. aureus* ATCC^®^ 25923^™^, which can be attributed to the fact that the bacteriocin activity is stronger against related bacterial strains, such as *L. monocytogenes* ATCC^®^ 19111^™^, than against other Gram-positive bacterial strains.

Next, we studied the capacity of the D13 strain to abolish biofilms formed by two pathogens in vitro. Namely, one of the approaches to suppressing the biofilms of pathogens in the food industry is the use of bacteriocins as an alternative to antibiotics [10]. Therefore, we evaluated the potential of strain D13 to inhibit biofilm formation. The CFS of strain *Lb. plantarum* D13 effectively inhibited the biofilms of both pathogens. Similar findings were reported by Zhao et al. [20], where the inhibition rate of *S. aureus* ATCC^®^ 25923^™^ biofilm formation was 22.60% and 52.39%, respectively, when plantaricin 827 was tested. These results provide the basis for the further investigation of the potential of the putative plantaricin from the CFS strain *Lb. plantarum* D13 as an antibiofilm agent.

In addition, the whole genome of *Lb. plantarum* D13 was sequenced and annotated. Genome mining analysis was performed to characterise the biosynthetic gene clusters encoding plantaricin. Comparative genome analyses of *Lb. plantarum* of different origins divided the strains into three lineages, A, B, and C, which differ in the presence of genes encoding bacteriocins, with PlnJK present in lineage A, while PlnEF is present in lineage A or C [21]. The production of some bacteriocins has been shown to be regulated by quorum sensing mediated by an induction peptide (IP), a membrane-bound sensor (histidine protein kinase; HPK), and a cytoplasmic response regulator (RR). This is consistent with our results, where a three-component quorum-sensing regulatory system was found in the genome of the D13 strain, together with genes encoding plantaricin A and the two-peptide plantaricins PlnEF and PlnJK, which were also found in other *Lb. plantarum* strains, namely LP965, EL3, L28, BL1, and SF9C [17,22,23]. Genes responsible for the production and putative biochemical function of these antimicrobials have also been deciphered, as have ABC transporters responsible for the secretion of bacteriocins and bacteriocin immunity proteins (PlnI, PlnP and PlnL, PlnM). *pln*Y and *pln*O, encoding biosynthesis proteins, and *pln*T, *pln*U, *pln*V, and *pln*W, encoding integral membrane proteins with the domain of the CAAX family of membrane-bound proteases, were identified. PlnW is a two-peptide bacteriocin that inhibits many strains of Gram-positive bacteria, whereas the function of the other three plantaricins has not yet been investigated. To fully understand the underlying mechanisms of bacteriocins, it is crucial to elucidate the structure, which will allow us to understand the molecular specificity of a given bacteriocin through structure–function analyses [17,24]. The predicted 3D homology structure of plantaricins EF and JK showed that they form amphiphilic α-helices with hydrophobic residues. This is consistent with the mode of action in which the bacteriocins of the class II bind to the membrane of the target cell via the N-terminal region due to their hydrophobic nature, while the C-terminal region forms an amphiphilic helical structure that allows them to penetrate the membrane, leading to the depolarisation and death of a susceptible microorganism. PlnE forms two α-helices, whereas PlnF forms one long helix. This is consistent with published findings that the two α-helix-like regions of PlnE are separated by a flexible GXXXG motif and are both amphiphilic, while the C-terminal part of the *pln*F helix is amphiphilic and the N-terminal part is polar. The hydrophobicity, hydrophobic moment, and net charge of plantaricins could be crucial for their antimicrobial activity. Accordingly, bacteriocins with more positive charge accelerate the binding of the α-helix to negatively charged phospholipids in the cell membrane through a stronger inter-charge interaction. Moreover, higher hydrophobicity improves penetration into the cell membrane, which accelerates bacterial cell disintegration, while the hydrophobic moment is used to quantify the amphipathicity of peptides [25].

Another functional feature of the use of *Lb. plantarum* and its plantaricin is the potential modulation of the intestinal microbiota and the resulting influence on intestinal disorders [26]. IBD is the most common intestinal disease, for which treatment with probiotic bacteria is often recommended [27]. IBD is associated with dysbiosis of the gut microbiota, which may lead to the development or exacerbation of the disease. Given its antimicrobial activity, we aimed to assess the capacity of *Lb. plantarum* D13 to restore the imbalanced intestinal microbiota of mice with DSS-induced colitis and potentially alleviate colitis symptoms. The most abundant phyla in the gut microbiota are *Firmicutes*, *Bacteroidetes*, *Actinobacteria*, and *Proteobacteria*, and changes in their distribution lead to dysbiosis. In our study, after the onset of colitis, dysbiosis was observed through a significant decrease in the relative abundance of *Bacteroidetes* and an increase in *Firmicutes* and *Actinobacteria*. The effect of *Lb. plantarum* D13 on the gut microbiota of DSS-induced mice was evident, not only in terms of the abundance of *Lb. plantarum* species but also in terms of the abundance of bacterial species associated with its beneficial effect. Namely, bacteria of the species *Allobaculum* produce butyric acid and have a positive effect on the expression of the protein ANGPTL4, which is a key regulator of lipid metabolism [28]. *Barnesiella intestinihominis* and *Barnesiella viscericola* are species associated with beneficial effects, such as protection against colitis. *Olsenella* spp. produce short-chain fatty acids, acetate and lactate [29], which play an important role in metabolic homeostasis and overall gut health. Conversely, the administration of *Lb. plantarum* D13 decreased the levels of *Candidatus Saccharimonas*, which causes inflammatory diseases of the intestinal mucosa, and *Ruminococcus flavefaciens*, which is associated with the risk of developing obesity and whose increased presence in the intestinal microbiome correlates with the development of IBS [30]. Our results imply that *Lb. plantarum* D13 contributes to the restoration of the gut microbiome after DSS-induced colitis, indicating the potential for further investigation as a probiotic strain for the prevention and treatment of colitis. 

## 4. Materials and Methods

### 4.1. Bacterial Strains and Cultivation Conditions

All *Lb. plantarum* strains and test microorganisms used in this study were deposited in the Culture Collection of the Laboratory for Antibiotic, Enzyme, Probiotic and Starter Culture Technologies, University of Zagreb Faculty of Food Technology and Biotechnology (CIM-FFTB), in de Man–Rogosa–Sharpe (MRS; Difco, Detroit, MI, USA) and Brain Heart Infusion (BHI; Biolife, Milano, Italy) broth, respectively, supplemented with 15% (*v/v*) glycerol (Sigma-Aldrich, Saint Louis, MO, USA), and were maintained as frozen stock at −80 °C (Eppendorf, Hamburg, Germany) (Table 1). *Lb. plantarum* strains were isolated from various autochthonous fermented food microbiomes in the Laboratory for Antibiotic, Enzyme, Probiotic and Starter Culture Technologies.

*S. aureus* ATCC^®^ 25923^™^ and *L. monocytogenes* ATCC^®^ 19111^™^ were used to test the antimicrobial activity of the selected *Lb. plantarum* strains. Prior to each experimental trial, the strains were transferred from the −80 °C stock culture and subcultured twice in an appropriate growth medium under the growth conditions listed in Table 1.

### 4.2. Characterisation of the Antimicrobial Activity and the pln Loci

The antimicrobial activity of the *Lb. plantarum* strains, against *S. aureus* ATCC^®^ 25923™ and *L. monocytogenes* ATCC^®^ 19111^™^, was determined by an agar spot test according to Butorac et al. [17]. The proteinaceous feature of the plantaricins was tested by treating the CFSs with 1 mg/mL proteinase K (Invitrogen, Carlsbad, CA, USA), pepsin (AppliChem, Darmstadt, Germany), and pancreatin (AppliChem, Darmstadt, Germany) for 2 h at 37 °C and heating the samples at 100 °C/30 min, with antimicrobial activity determined by the agar spot test method, against *S. aureus* ATCC^®^ 25923^™^ and *L. monocytogenes* ATCC^®^ 19111^™^, according to Butorac et al. [17]. PCR screening of plantaricin structural genes was performed according to Butorac et al. [17]. The PCR products were sequenced by Macrogen (Amsterdam, The Netherlands). Sequences were aligned to the NCBI nt database using the BLASTn algorithm v. 2.11.0 (https://blast.ncbi.nlm.nih.gov/Blast.cgi, accessed on 2 May 2023) [31]. The potential to induce plantaricin biosynthesis of *Lb. plantarum* D13 was investigated by performing cocultivation experiments according to Butorac et al. [17]. CFSs were obtained by centrifugation of the overnight grown medium at 13,000 rpm for 10 min, and the supernatants were filtered through a Millipore filter (Sigma-Aldrich, Saint Louis, MO, USA) with a pore size of 0.22 µm. The pH values of CFSs were measured using a pH meter (SI Analytics, Mainz, Germany) and titratable acidity by titration with 0.1 M NaOH (Kemika, Zagreb, Croatia). 

### 4.3. Inhibition of Biofilm Formation

The influence of the CFS of the selected *Lb. plantarum* D13 strain on the biofilm formation of *L. monocytogenes* ATCC^®^ 19111^™^ and *S. aureus* ATCC^®^ 25923^™^ was performed in vitro as described by Bazargani and Rohloff [32], with slight modifications. The foodborne pathogenic strains tested were grown overnight in BHI broth and the suspension was adjusted to correspond to a McFarland turbidity of 0.5 (Biolife, Milan, Italy). Then, 100 µL of the bacterial suspension was inoculated onto a sterile 96-well microtitre plate (Corning Inc., Corning, New York, NY, USA). The bacterial suspensions were treated with 100 µL of the CFS of the strain *Lb. plantarum* D13. Pathogen growth in the BHI medium served as a control. The inoculated microtitre plate was incubated at 37 °C for 24 h under aerobic conditions. After incubation, the medium was removed and the wells were washed three times with phosphate-buffered saline (pH = 7.4) (FFTB, Zagreb, Croatia). Then, 100 µL of crystal violet (Thermo Fisher Scientific, Waltham, MA, USA) was added to each well and they were incubated at room temperature for 30 min. The dye was removed and then the wells were washed 5 times with sterile distilled water and decolorised with 200 µL of 96% ethanol (Merck Milipore, Darmstadt, Germany) for 10 min at room temperature. Finally, 100 µL of the ethanol–dye suspension from each sample was transferred to a different 96-well microplate where the optical density (OD) was measured at 540 nm using the Infinite F Plex microplate reader (Tecan, Grödig, Austria). The inhibition ratio was calculated as follows:I% = (1 − A/A_0_) × 100%
where A and A_0_ represent the absorbance of the CFS-treated bacterial suspension and the bacterial suspension, respectively.

### 4.4. Genome Sequencing and Analysis of the pln Loci

The strain *Lb. plantarum* D13, isolated from smoked fresh cheese, was identified by whole genome sequencing (WGS) at IGA Technology Services Srl (Udine, Italy). Genomic DNA was extracted using the Maxwell^®^ DNA Cell Kit in the Maxwell^®^ 16 Research System (Promega, Madison, WI, USA) according to the manufacturer’s recommendations. Sequencing of 16S rRNA variable regions V3–V4 was performed with the MiSeq 2500 instrument from Illumina (Illumina, San Diego, CA, USA) using the paired-end approach as described in Banić et al. [15]. The whole genome sequences of strain D13 were deposited in the NCBI database under BioProject PRJNA388578 (v. 1.0) with BioSample accession number SAMN07179270. CASAVA v. 1.8.2 was used for data processing. The annotation, distribution, and categorisation of sequenced genes were performed using the Rapid Annotation using Subsystem Technologies (RAST) SEED viewer (http://rast.nmpdr.org/, accessed on 15 February 2023) [33]. The identification of plantaricin-related genes and subsequent assigned functions, shown as clusters, was performed using Genome Browser. Three-dimensional structure prediction of the discovered plantaricins was performed using the AlphaFold Protein Structure Database (https://alphafold.ebi.ac.uk/, accessed on 4 April 2023) [34,35]. The HeliQuest web server was used to calculate the physicochemical properties and amino acid compositions of the α-helices in the discovered plantaricins (https://heliquest.ipmc.cnrs.fr/, accessed on 20 April 2023) [36].

### 4.5. In Vivo Animal Trial

#### 4.5.1. Experimental Animals

Male and female C57BL/6 mice, aged two to three months and weighing 20 to 30 g, obtained from the Department of Animal Physiology, Faculty of Science, University of Zagreb, were used for this study. During the experiment, the animals were kept under standard conditions (temperature 25 (±3) °C, relative humidity 55 (±10)%, and 12 h of light and 12 h of darkness) in cages, separated by sex and feeding conditions. Prior to the experiment, the mice were fed a standard laboratory diet, and water was provided ad libitum. The food pellets were a certified standard diet for mice: 4RF21 (Mucedola, Settimo Milanese, Italy; batch no. 238603, shape 12 mm). The maintenance, housing, handling, and care of all experimental animals were performed according to the guidelines in force in the Republic of Croatia (Animal Welfare Act, OG 19/1999) and according to the Guide for the Care and Use of Laboratory Animals, DHHS Publ. # (NIH) 86-23. All guidelines are in accordance with the internationally accepted principles for the use and care of laboratory animals as found in the European Community guidelines (EEC Directive of 1986; 86/609/EEC). The Ethics Committee of the Faculty of Science (University of Zagreb, Croatia) approved the study (approval code: 251-58-10617-19-285).

#### 4.5.2. DSS-Induced Colitis and Administration of *Lb. plantarum* D13 Strain

C57BL/6 mice received 3% DSS (36,000–50,000 Da) (MP Biomedicals, Santa Ana, CA, USA) in sterile water for 5 days to establish a colitis mouse model, followed by the administration of *Lb. plantarum* D13. Strain D13 was administered intragastrically daily for 6 consecutive days with 500 µL of a bacterial suspension suspended in saline at a final concentration of 10^9^ CFU/mL. Faecal samples were collected before treatment with DSS, after treatment/before administration, on day 3 and day 6 of administration, and 5 days after the administration of *Lb. plantarum* D13. A healthy mouse model administrated strain D13 served as a control. DNA from the mice faecal samples collected at specific time points was isolated and analysed according to Butorac et al. [17]. In brief, total DNA from faecal mice samples was extracted using a Maxwell^®^ DNA Tissue Kit (Promega, Madison, WI, USA) according to the manufacturer’s instructions. Samples were sequenced at Molecular Research LP (MRDNA, Shallowater, TX, USA) with the Illumina MiSeq platform using primers 341F (5′-CCTACGGGNGGCWGCAG-3′) and 518R (5′-ATTACCGCGGCTGCTGG-3′). No side effects were reported after administration. To assess whether colitis was induced, the DAI was determined separately for each animal using the measured parameters that corresponded to the clinical picture of ulcerative colitis in humans [37]. The following parameters were used for calculation: (a) weight loss (0 = none, 1 = 1–3% weight loss, 2 = 3–6% weight loss, 3 = 6–9% weight loss, and 4 = >9% weight loss); (b) stool consistency/diarrhoea (0 = normal, 2 = loose stool, 4 = watery diarrhoea); (c) bleeding (0 = no bleeding, 2 = mild bleeding, 4 = severe bleeding). The DAI is calculated as a total score, i.e., the sum of weight loss, diarrhoea, and bleeding, which can result in a DAI value from 0 (unchanged) to 12 (severe colitis). To assess histological alterations in the distal colon, the tissue samples were fixed in 4% paraformaldehyde (Gram Mol, Zagreb, Croatia), embedded in paraffin (Sigma-Aldrich Chemie GmbH, Taufkirchen, Germany), and cut into 5-μm-thick sections. The slides were then stained with haematoxylin (Merck, Darmstadt, Germany) and eosin (Acros Organics, Geel, Belgium) (HE) and analysed by a pathologist on a Zeiss Axio Star microscope (Carl Zeiss Microscopy GmbH, Jena, Germany). The pathohistological assessment was performed according to the work by Koelink et al. [38] and included the following parameters: the presence of ulcerations, inflammatory cells (such as neutrophils, macrophages, lymphocytes, and plasma cells), signs of oedema, crypt loss, surface epithelial cell hyperplasia, goblet cell reduction, and signs of epithelial regeneration. The evaluation of the degree of ulcerative lesions was performed using the mouse colitis histology index (MCHI), which includes eight histological components: inflammatory infiltrate, goblet cell loss, hyperplasia, crypt density, muscle thickness, submucosal infiltration, ulcerations, and crypt abscesses (all categorised from 0 to 3). The histopathological score was calculated based on the sum of the eight scores, ranging from 0 to 24: MCHI = 1 × goblet cell loss [four categories] + 2 × crypt density [three categories] + 2 × hyperplasia [four categories] + 3 × submucosal infiltrate [four categories].

### 4.6. Statistical Analysis

All experiments were performed in triplicate and results were reported as the mean of three independent trials ± standard deviation (SD). Statistical significance was appraised by analysis of variance (ANOVA). Pairwise differences between the mean values of groups were determined using Tukey’s honestly significant difference (HSD) test for post-analysis of variance pairwise comparisons (https://www.statskingdom.com/index.html, accessed on 1 September 2023) (Statistics Kingdom, 2017). Differences between groups were considered significant at *p* < 0.05. Microbiota diversity analysis was performed using the PAleontological STatistics (PAST) software, v. 4.13 [39]. Images were created using GraphPad Prism v.9.4.1 (GraphPad Software, San Diego, CA, USA).

## 5. Conclusions

In this study, we show that plantarcin-producing *Lb. plantarum* D13 has the probiotic potential to exclude *S. aureus* and *L. monocytogenes* and modulate the gut microbiota of healthy or DSS-induced colitis in mice by balancing the ratio of beneficial and commensal species. The biomodulation capacity is supported by the analysis of the specific antimicrobial activity and the characterisation of the putative D13 plantaricin by identifying the plantaricin biosynthesis cluster, predicting its 3D structure, and demonstrating its antibiofilm properties. Further studies are needed to decipher the underlying mechanisms related to the selected D13 strain and to translate these in vivo findings into precise molecular mechanisms. However, taken together, these results encourage future studies to characterise the use of plantaricin-producing *Lb. plantarum* D13 as a biomodulation agent to potentially combat colitis.

## Figures and Tables

**Figure 1 ijms-24-15322-f001:**
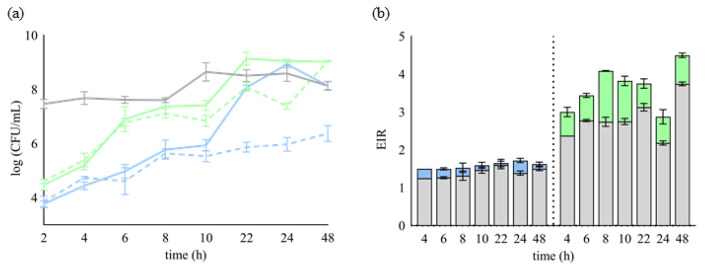
Growth inhibition (log (CFU/mL)) of *L. monocytogenes* ATCC^®^ 19111™ (− −) and *S. aureus* ATCC^®^ 25923™ (− −) during 48 h of co-cultivation with *Lb. plantarum* D13. Growth curves of *Lb. plantarum* D13 (−), *L. monocytogenes* ATCC^®^ 19111™ (−), and *S. aureus* ATCC^®^ 25923™ (−) alone (**a**). The corresponding values of the effective inhibition ratio (EIR) for the test microorganisms. The lower grey part of the bars represents the EIR of *Lb. plantarum* D13 alone (█), and the upper part of the bar represents the EIR of *Lb. plantarum* D13 towards *L. monocytogenes* ATCC^®^ 19111™ (█) and *S. aureus* ATCC^®^ 25923™ (█), during 48 h of cocultivation (**b**). Results are given as mean ± SD of three independent experiments.

**Figure 2 ijms-24-15322-f002:**
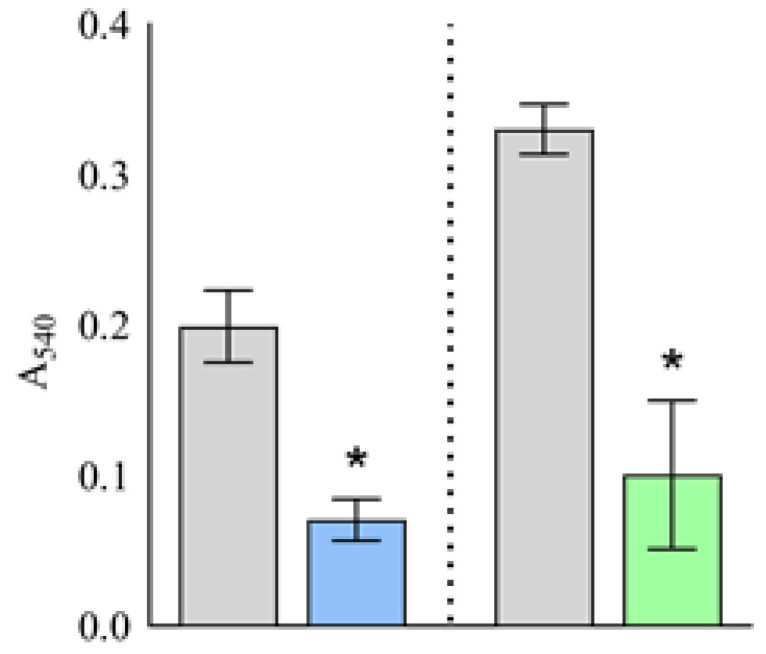
Effects of cell-free supernatant (CFS) of *Lb. plantarum* D13 on biofilm formation (A_540_) of *L. monocytogenes* ATCC^®^ 19111^™^ (█) and *S. aureus* ATCC^®^ 25923^™^ (█). * Significant difference (*p* < 0.001) from the untreated control of the respective test microorganisms (█).

**Figure 3 ijms-24-15322-f003:**
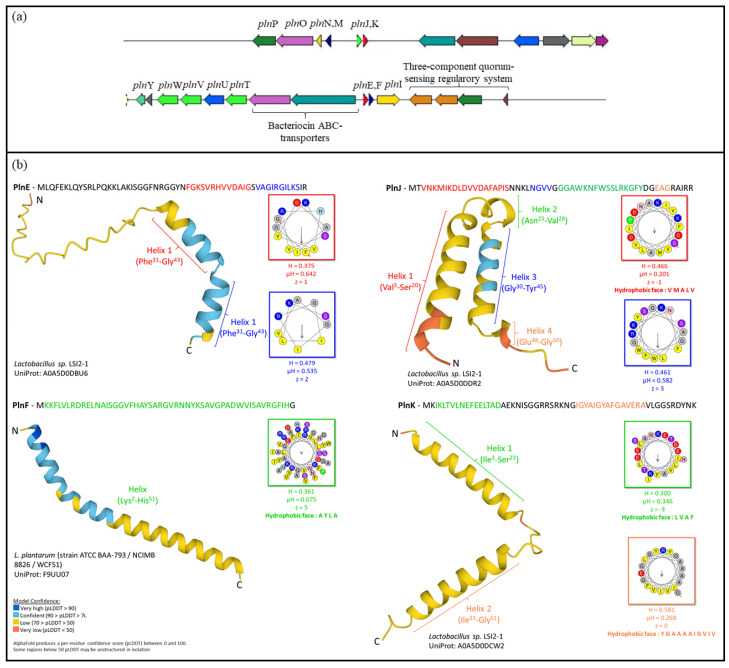
Mosaic *pln* gene cluster of *Lb. plantarum* D13 strain (**a**). The 3D homology structure prediction of PlnE, PlnF, PlnJ, and PlnK of strain D13 using AlphaFold Protein Structure Database. The source microorganisms with a sequence similarity threshold of 100% and the respective UniProt ID are indicated, and the helical-wheel representation of regions that may form an α-helix were analysed with HeliQuest, where a one-letter code is used for one amino acid; yellow—hydrophobic residues; purple—serine and threonine residues; dark blue—basic residues; red—acidic residues; pink—asparagine and glutamine residues; grey—alanine and glycine residues; light blue—histidine residues; green—proline residues. H—hydrophobicity; μH—hydrophobic moment; z—net charge (calculated at pH = 7.4, assuming that histidine is neutral and that the N-terminal amino group and the C-terminal carboxyl group of the sequence are uncharged) (**b**).

**Figure 4 ijms-24-15322-f004:**
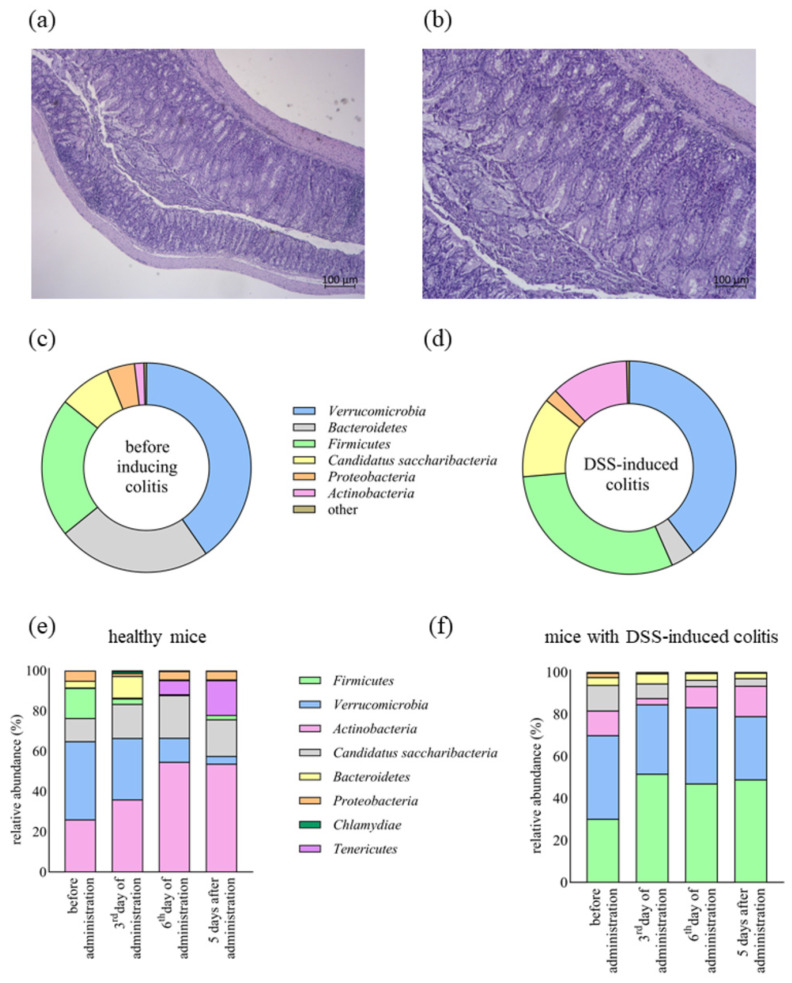
In situ dysplasia of colonic mucosa in mice with DSS -induced colitis, with (**a**) and without (**b**) application of *Lb. plantarum* D13. Analysis of the composition of the gut microbiota at the phylum level before (**c**) and after (**d**) DSS-induced colitis in mice, during different phases of administration of *Lb. plantarum* D13 (3rd and 6th day), and on 5th day after the end of administration in healthy mice (**e**) and mice with DSS-induced colitis (**f**). Phyla with an average prevalence of less than 1% are classified as other.

**Figure 5 ijms-24-15322-f005:**
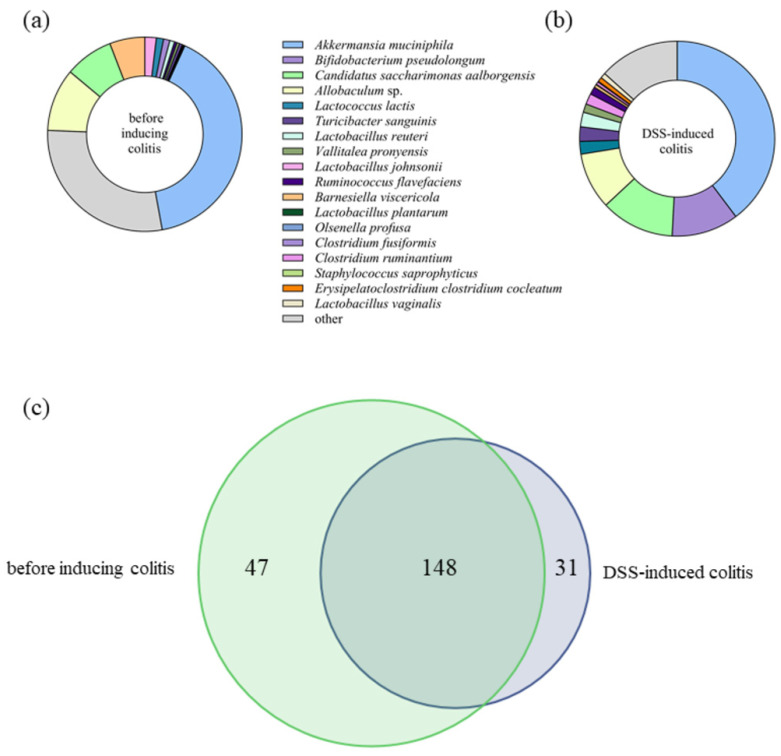
The relative abundance (%) of bacterial species of the gut microbiota before (**a**) and after DSS-induced colitis in mice (**b**). Venn diagram of shared and unique bacterial operational taxonomic units (OTUs) among groups before and after DSS-induced colitis (**c**). Species with an average representation of less than 1% are classified as other.

**Figure 6 ijms-24-15322-f006:**
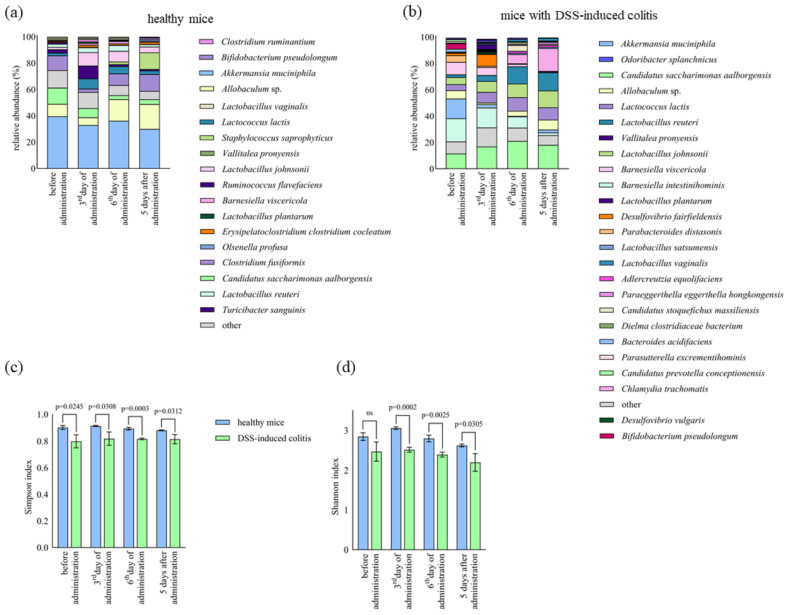
Relative abundance (%) of species before, during (3rd and 6th day), and after (5 day) administration of *Lb. plantarum* D13 in healthy mice (**a**) and mice with DSS-induced colitis (**b**). Alpha diversity of operational taxonomic units (OTUs) using Simpson (**c**) and Shannon indices (**d**), between healthy and DSS-induced mice during administration of *Lb. plantarum* D13. Species with an average representation of less than 1% are classified as other. ns—not significant.

**Table 1 ijms-24-15322-t001:** Bacterial strains, cultivation conditions, and isolation sources used in this study.

Bacterial Strain	Cultivation Conditions	Source
*Lb. plantarum* D4	MRS, 37 °C, microaerophilic	Smoked fresh cheese
*Lb. plantarum* D5	MRS, 37 °C, microaerophilic	Smoked fresh cheese
*Lb. plantarum* D7	MRS, 37 °C, microaerophilic	Smoked fresh cheese
*Lb. plantarum* D13	MRS, 37 °C, microaerophilic	Smoked fresh cheese
*Lb. plantarum* M4	MRS, 37 °C, microaerophilic	Dried fresh cheese
*Lb. plantarum* M5	MRS, 37 °C, microaerophilic	Dried fresh cheese
*Lb. plantarum* MA2	MRS, 37 °C, microaerophilic	Dried fresh cheese
*Lb. plantarum* MA3	MRS, 37 °C, microaerophilic	Dried fresh cheese
*Lb. plantarum* SF15C	MRS, 37 °C, microaerophilic	Sauerkraut
*Lb. plantarum* ZG1C	MRS, 37 °C, microaerophilic	Sauerkraut
*Lb. plantarum* M92C	MRS, 37 °C, microaerophilic	Fermented milk
*Lb. plantarum* L4	MRS, 37 °C, microaerophilic	Silage
*S. aureus* ATCC^®^ 25923^™^	BHI, 37 °C, aerobic	ATCC ^1^
*L. monocytogenes* ATCC^®^ 19111^™^	BHI, 37 °C, aerobic	ATCC ^1^

^1^ ATCC—American Type Culture Collection.

## Data Availability

Not applicable.

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
