# Peer review of "Modulation of the Gut Microbiota by the Plantaricin-Producing Lactiplantibacillus plantarum D13, Analysed in the DSS-Induced Colitis Mouse Model"

_ijms, 2023, doi:10.3390/ijms242015322_

Round 1

Reviewer 1 Report

The paper by Butorac and co-workers presents a multi-level analysis of the probiotic potential of a Lb. plantarum strain, focussing on its capacity to produce plantaricin and to modulate gut microbiota composition. The paper is well written, protocols and results obtained clearly presented, and the contribution to the field is evident.

I only have minor suggestions:

-        Line 82: I would rephrase as “…all strains of Lb. plantarum displayed strong anti-staphylococcal inhibition..”

-        Lines 125-132 and figure 1a: while the inhibitory effect of Lb. plantarum on the growth of L. monocytogenes is evident both at 24 and 48h of co-cultivation, on S. aureus the difference observed at 24 h is lost after 48h. Could authors better comment on this result?

-        Lines 133-138: I find these few sentences rather unclear. I suggest to rephrase them to make it more evident how it was possible to conclude that plantaricin -and not lactic acid- is the reason for the inhibitory effect.

-        Paragraph 2.2: the effect on the inhibition of biofilm formation is clear. However, do authors have evidences about the effect of Lb. plantarum D3 on the eradication of an already formed biofilm?

Author Response

Manuscript ID: ijms-2671363

Title: Modulation of the gut microbiota by the plantaricin-producing Lactiplantibacillus plantarum D13, analysed in the DSS-induced colitis mouse model

Reviewer 1

We would like to thank you for handling our manuscript through the peer review process. We greatly appreciate the suggestions received from you and would like to thank you for investing your time in reviewing our manuscript. Please see below, a point-by-point response to the comments:

The paper by Butorac and co-workers presents a multi-level analysis of the probiotic potential of a Lb. plantarum strain, focussing on its capacity to produce plantaricin and to modulate gut microbiota composition. The paper is well written, protocols and results obtained clearly presented, and the contribution to the field is evident.

I only have minor suggestions:

Line 82: I would rephrase as “…all strains of Lb. plantarum displayed strong anti-staphylococcal inhibition..”

Response: Than you for your suggestion, the Line 82 was rephrased.

-        Lines 125-132 and figure 1a: while the inhibitory effect of Lb. plantarum on the growth of L. monocytogenes is evident both at 24 and 48h of co-cultivation, on S. aureus the difference observed at 24 h is lost after 48h. Could authors better comment on this result?

Response: To answer your question, these results have been further commented on in the Discussion section. Please see lines 361-364:

„Antibacterial activity was weaker against S. aureus ATCC® 25923, which can be attributed to the fact that bacteriocin activity is stronger against related bacterial strains, such as L. monocytogenes ATCC® 19111 than against other Gram-positive bacterial strains.“

-        Lines 133-138: I find these few sentences rather unclear. I suggest to rephrase them to make it more evident how it was possible to conclude that plantaricin -and not lactic acid- is the reason for the inhibitory effect.

Response: Thank you for this valuable comment. For clarification the sentence was rephrased, please see lines 124-129:

„The antimicrobial activity of the acids produced by the metabolism of the Lb. plantarum D13 strain was eliminated by monitoring the pH values during the cocultivation, which corresponded to the values measured in the monoculture of the test microorganisms, and additionally by testing their inhibition using the determined lactic acid concentrations in coculture (8.1 and 6.3 g/L with L. monocytogenes ATCC® 19111 and S. aureus ATCC® 25923, respectively) at the end of the experiment. “

-        Paragraph 2.2: the effect on the inhibition of biofilm formation is clear. However, do authors have evidences about the effect of Lb. plantarum D3 on the eradication of an already formed biofilm?

Response: Thank you very much for this valuable comment, based on which the revised text is as follows in lines 163-166:

„Given the demonstrated anti-biofilm and antimicrobial activities of the Lb. plantarum D13, we can suggest its potential ability to eradicate an already formed biofilm in the microenvironment against susceptible bacterial species (Figures 1 and 2).“

Reviewer 2 Report

In this study, Butorac et al. investigated the antistaphylococcal and antilisterial activities of Lactiplantibacillus plantarum D13, as well as its effects on DSS-induced colitis mice. The manuscript is well-written and the topic of this study is interesting, but I have a few concerns. Here are some comments on this paper:

1.       Regarding cell-free supernatants (CFSs) of Lb. plantarum, could the authors provide more detailed information, such as preparation methods, composition and concentration?  As mentioned by the authors in the manuscript lines 133-139 lactic acid has antimicrobial activity. If lactic acid is in the CFSs, how to determine whether the antimicrobial property is plantaricin or lactic acid, even both? I consider the concentrations of lactic acid to be very important and suggest that authors provide this result. Moreover, lines 137-139 “Therefore, it can be speculated 137 that the inhibitory effect of Lb. plantarum D13 during cocultivation is a possible 138 consequence of plantaricin production” is confusing, how causality is established?

2.       The authors investigated possible plantaricin-related genes of Lb. plantarum D13 genome in section 2.3. It is advisable to verify their functions by in vitro expression and associate the genes/proteins with antimicrobial activity functions.

3.       Line 217 “Metagenomic” should be 16S rRNA gene. Line 256 “colon” should be “fecal samples”.

4.       Figure 4 a hematoxylin and eosin should be evaluated through scoring.

5.       From Figure 6 c and d, we could find that Lb. plantarum D13 had no effect on alpha diversity in the DSS-induced group mice. Could authors give any explanation for this?

6.       It is proposed that the authors specify the methods of detection and analysis of intestinal microbiome in section 4.5.2.

Author Response

Manuscript ID: ijms-2671363

Title: Modulation of the gut microbiota by the plantaricin-producing Lactiplantibacillus plantarum D13, analysed in the DSS-induced colitis mouse model

Reviewer 2

I would like to thank you for handling our manuscript through the peer review process. We greatly appreciate the attention and suggestions we received from you and would like to thank you for investing your valuable time in reviewing our manuscript. Regarding you comments our responses are given as follows:

In this study, Butorac et al. investigated the antistaphylococcal and antilisterial activities of Lactiplantibacillus plantarum D13, as well as its effects on DSS-induced colitis mice. The manuscript is well-written and the topic of this study is interesting, but I have a few concerns. Here are some comments on this paper:

  1. Regarding cell-free supernatants (CFSs) of Lb. plantarum, could the authors provide more detailed information, such as preparation methods, composition and concentration? As mentioned by the authors in the manuscript lines 133-139 lactic acid has antimicrobial activity. If lactic acid is in the CFSs, how to determine whether the antimicrobial property is plantaricin or lactic acid, even both? I consider the concentrations of lactic acid to be very important and suggest that authors provide this result. Moreover, lines 137-139 “Therefore, it can be speculated 137 that the inhibitory effect of Lb. plantarum D13 during cocultivation is a possible 138 consequence of plantaricin production” is confusing, how causality is established?

Response: Thank you very much for these valuable comments. For clarification, the praparation of CFSs was added in the Materials and Methods section (subsection 4.2. Characterisations of the antimicrobial activity and the pln loci), please see lines 469-473. The part regarding the lactic acid was rephrased and the concentrations of the lactic acid was added. Please see lines 124-132.

Lines 469-473: „CFSs were obtained by centrifugation of the overnight grown medium at 13000 rpm for 10 min, and the supernatants were filtered through a Milipore filter (Sigma-Aldrich, Saint Louis, MO, USA) with a pore size of 0.22 µm. The pH values of  CFSs  were measured using a pH meter (SI Analytics, Mainz, Germany) and titratable acidity by titration with 0.1 M NaOH (Kemika, Zagreb, Croatia).“

Lines 124-132: „The antimicrobial activity of the acids produced by the metabolism of the Lb. plantarum D13 strain was elimanted by monitoring the pH values during the coculture which corresponded to the values measured in the monoculture of the test microorganisms, and additionally by testing their inhibition using the determined lactic acid concentrations in coculture (8.1 and 6.3 g/L with L. monocytogenes ATCC® 19111 and S. aureus ATCC® 25923, respectively) at the end of the experiment. Considering the monitored pH values and the stronger antimicrobial activity of CFSs compared to lactic acid, it can be speculated that the inhibitory effect of the Lb. plantarum D13 strain during cocultivation is the result of possible plantaricin activity.“

  1. The authors investigated possible plantaricin-related genes of Lb. plantarum D13 genome in section 2.3. It is advisable to verify their functions by in vitro expression and associate the genes/proteins with antimicrobial activity functions.

Response: Thank you for pointing this out. Accordingly, the following sentence was introduced to comment the associated activity of plantaricins EF and JK . Please see lines 193-195.

Lines 193-195: „Since their structure shares 100% amino acid identity with the respective peptides of  other strains with characterised plantaricin antimicrobial activity, it is presumed that their function could be similar.“

  1. Line 217 “Metagenomic” should be 16S rRNA gene. Line 256 “colon” should be “fecal samples”.

Response: Thank you for this notes, the above mentioned was corrected.

  1. Figure 4 a hematoxylin and eosin should be evaluated through scoring.

Response: Thank you very much for this valuable suggestion. As you suggested, hematoxylin and eosin were evaluated through scoring, and we replaced Figure 4a with a new picture for clarification. Please refer to lines 233-247.

Histological analysis and scoring were additionally described in the Materials and Methods section (subsection 4.5.2. DSS-induced colitis and administration of Lb. plantarum D13 strain). Also please refer to lines 560-577.
